# Occupational Health and Safety Receptivity towards Clinical Innovations That Can Benefit Workplace Mental Health Programs: Anxiety and Hypnotherapy Trends

**DOI:** 10.3390/ijerph19137735

**Published:** 2022-06-24

**Authors:** Petrina Coventry

**Affiliations:** School of Management, Asian Institute of Technology, Klong Luang, Pathumthani 12120, Thailand; petrina.coventry@adelaide.edu.au

**Keywords:** anxiety, occupational health and safety, workplace productivity, hypnotherapy

## Abstract

Anxiety is one of the most common mental health conditions experienced by people in Australia during their working years according to the Australian Bureau of Statistics (ABS) and employers recognising that mental health impacts their organisation’s bottom line are increasingly interested in programs to promote better mental health, well-being, and productivity. Beyond management concern for productivity, statutory protection is necessary to safeguard mental health, under the Australian Occupational Health and Safety (OHS) Act, organisations have a duty to eliminate or minimise risks to psychological (mental) health by designing effective workplace systems and Occupational Health and Safety (OHS) practitioners are central to the design and responsibility in managing these systems. Despite literature indicating the benefits of OHS workplace mental health initiatives, such as improved overall health, reduced absenteeism, increased job satisfaction and morale, there remains a lack of empirical research around program measurement, and their effectiveness in this area has been brought into question. The OHS function is interested in improving research around the relationship and connection between work performance and mental health but: there are few studies regarding performance outcomes of mental health OHS services within management journals and insufficient information around the prevalence of psychological morbidity in the workplace and its impact. The purpose of this study was to assess OHS perception regarding anxiety and reveal perceptions and receptivity towards alternative therapies and solutions being used in clinical practice to deal with anxiety such as cognitive behavioural therapy (CBT) with a focus on Hypnotherapy.

## 1. Introduction

### 1.1. Anxiety Is a Normal Everyday Condition: Diagnosis Not Required

One in thirteen people suffer from anxiety [1] and anxiety is recognised as a cognitive and emotional state not driven by conscious logic. But with the root causes existing in the subconscious brain [2]. 

Triggers exist within the limbic region of the brain, sometimes equated to the subconscious hard drive. When confronted with emotional stressors, this system is more immediate in terms of the time response than the conscious mind [3,4] and can overtake the executive, or rational, functioning of the brain and individual. As the state of anxiety is subconscious in nature, in practice, the treatment needs to be subconscious to address the issue [5]. 

Emotion and anxiousness are highly personalised, varying widely by individual, and these emotions can occur as the result of a stimulus that triggers automatic responses, which are normally fear, panic, and worry. When emotions or anxiousness are extreme, subconscious structures can stimulate the same repeated automatic responses (reactions) until neural patterns or behaviours are subconsciously modified [6]. 

Understanding the subconscious and tacit knowledge connecting anxiety and performance [7,8,9,10,11,12] This theory highlights how anxiety negatively affects functioning by decreasing attentional control over a performed action impacting the processing and storage capacity of working memory and reducing the resources available for a given task and increasing attention to threat-related stimuli [8]. In the workplace, this can present in a number of ways to impact performance.

### 1.2. Anxiety in the Workplace: It Comes to Work and Can Be Exacerbated at Work

Anxiety is an ongoing emotional state; it is not “nine to five” nor only borne out of the work environment. General everyday life can trigger and affect emotions and feelings of anxiety, and individuals bring their concerns, emotions, anxiety and worries to work. Emotions influence how people work and underpin behaviours in organisations [13,14,15]. Emotional and anxious states can be exacerbated within the workplace where the risk of failing at tasks, and psychological drives to find security, prevail. 

Normal anxiety, or feelings of anxiousness, are quite common and do not require a medical diagnosis but can still be problematic [16,17].

When anxiety escalates to a problematic, uncontrollable level, it can interfere with an individual’s ability to effectively function socially, including within their work environment, consequently impacting work performance [18,19]. 

### 1.3. Productivity Impact

Without effective management, anxiety can seriously impact individual employee productivity with potential major implications for overall organisational performance [19]. Employees experiencing anxiety suffer more than double the risk of poor work performance [18] through characteristics of decreased patience, inability to multitask, difficulty in making decisions, lack of concentration, a slowdown in task delivery, and overall employee engagement levels [19]. These characteristics can negatively impact organisational productivity through relationships with co-employees and peers (51%), quality of work (50%), and relationships with superiors (43%) [20]. Rising long-term absences, work injury claims, and incapacity benefits have all been attributed to mental disorders including anxiety [21].

### 1.4. Occupational Health and Safety (OHS) Responsibility

Occupational Health and Safety (OHS) legislation often specifies the need for employers to act against psychosocial hazards [22], and spending on OHS mental health and well-being programs exceeds USD 50 billion globally, with expectations that it will grow by a further 7% annually [23]. However, with past emphasis on physical safety within OHS programs, often at the expense of psychological safety research and innovation, there is more work to be done determining the real impact and success of these programs, and to address questions regarding OHS effectiveness to manage psychological safety issues such as anxiety [24,25,26]. 

While extant research tends to focus on alleviation of symptoms and risk factors associated with mental disorders, less emphasis has been placed on gathering evidence on how anxiety affects work performance and how OHS service offerings around treating anxiety have resulted in improvement in work performance. Due to the shortage of research in this area [10,27] looking to the field of clinical studies may provide assistance.

### 1.5. Leaders in the Field

The possibility of subconscious CBT clinical trends can help to determine if there may be opportunities for workplace solutions for OHS. 

Anxiety, anxiousness, and mental health conditions are increasingly presenting to clinicians as a cause for concern, and there are innovative measures being developed and assessed to remediate the issue [28,29,30].

Anxiety may be successfully managed using conservative management, including practitioner-directed and self-help techniques, referred to as sub-conscious cognitive therapies. These therapies are based on in-depth understanding of the mechanism that leads to states of anxiety, and accepted clinical therapies include mindfulness (relaxation and awareness), and, increasingly, hypnotherapy. 

### 1.6. Hypnotherapy—What It Is, and Isn’t

Hypnotherapy is recommended for a number of conditions, such as pain management, behaviour management, and anxiety [6,31,32,33,34].

Hypnotherapy is a therapeutic modality in which hypnosis is used [6,31,33,35,36], with the clinical efficacy of the technique for anxiety management increasingly being objectively validated through neural imaging using functional magnetic resonance imaging [37,38,39,40]. 

The National Health Service in the UK defines hypnotherapy as “a type of complementary therapy that uses hypnosis, which is an altered state of consciousness” and the American Psychological Association (2017) defines hypnotherapy as a “state of consciousness involving focused attention and reduced peripheral awareness characterised by an enhanced capacity for response to suggestion”. 

Some suggest that hypnotherapy is similar to mindfulness in that it is based on attention and the power of suggestion through openness to acceptance to new ideas [41]. The two practices differ. Mindfulness is a conscious state, whereas hypnotherapy relies on tapping into the subconscious mind. It is important to consider the subconscious nature of anxiety. Empirical data indicates that combining hypnosis and mindfulness may result in greater therapeutic outcomes, compared to either modality used in isolation [31,42,43,44]. 

Hypnotherapy offers a subconscious cognitive solution and clinicians appear to be progressing in understanding this [2]. 

The technological advancement of neural imaging techniques is assisting the development of clinical practice in this field by improving conclusions around how innovative and alternative solutions such as CBT and hypnotherapy help alleviate anxiety.

### 1.7. Hypnotherapy Inclusion in OHS Services

Despite the increased acceptance of this field in clinical practice, the formal use and reference to hypnotherapy within workplace settings appear to be noticeably absent, and there is a dearth of OHS research represented in the literature.

## 2. Research Aims

The aim of this study, conducted in Australia, was to gather evidence around perceptions, barriers, and opportunities presented as to whether: hypnotherapy:Hypnotherapy is considered an effective cognitive behavioural therapy (CBT) to deal with anxiety in clinical practice,has a place in workplace OHS programs to manage anxiety; andwhat barriers towards receptivity may need to be overcome by OHS before being considered a workplace service?

## 3. Methods and Results

A mixed methods study was conducted in partnership with the Australian Institute of Health and Safety (AIHS), the peak body representing OHS professionals within the industry and responsible for accredited OHS training and practitioner registration. Human Research Ethics approval is important within academic studies in Australia and ethics approval was sought and gained under the Australian university research protocols. This was considered low risk research.

The study included: 1. a literature review; 2. quantitative research, using an online self-administered survey; and 3. qualitative research through the conduct of a focus group.

### 3.1. Literature Review

A literature review (Figure 1) was conducted to investigate evidence regarding the efficacy of alternative therapies that deal with subconscious matters and neural patterns including anxiety. This included CBT and hypnotherapy within clinical settings and within OHS.

The strategy used was a computer-assisted literature search conducted using the following databases: EBSCO Host including Academic Search Premier, PsycArticles, Psychology and Behavioural Sciences Collection, PsycBooks, PsycInfo, Sage Psychology Journals, Proquest Psychology Journals, ScienceDirect Social and Behavioural Sciences, and the Cochrane Central Register of Controlled Trials.

The study selection included:Meta-analysis of randomised controlled trialsRandomised controlled trialsControlled studies without randomisationDescriptive studies such as comparative studies, correlation-based studies, or case control studies, expert committee reports or opinions, clinical experience, or respected authority, or both.

After two levels of filtering, articles remained that met the inclusion criteria, which reviewed the acceptance of the use of hypnotherapy in clinical and workplace settings.

The literature review highlighted an issue around the lack of studies on OHS performance outcomes covered in management journals [24,25,27,45,46,47]. There is little evidence and exploration around the field of anxiety and the use of alternative therapies in OHS including hypnotherapy.

#### Literature Review Findings around Clinical Trends

In the past, first-line treatment options for anxiety were reliant on the use of pharmaceutical drugs including selective serotonin uptake inhibitors (SSRIs), in addition to antidepressants and benzodiazepines, noting that these are typically only recommended for acute anxiety reactions, and not for the treatment of everyday anxiousness.

Non-pharmacologic interventions in the form of psychotherapy are increasingly used to treat anxiety and anxiousness with an upsurge of interest in psychotherapy including CBT and hypnotherapy.

Psychotherapy programs that address symptoms and treatment requirements of individual anxiety disorders can include elements of desensitisation, exposure, and cognitive restructuring, allowing the participant to demonstrate new learning and developing new neural patterns in real-life situations [48]. 

Hypnotherapy uses neural path connections between the conscious and the subconscious mind, enabling reframing and release of previously pre-programmed automatic responses, replacing them with more helpful adaptive ways to think. This can reduce anxiousness and result in beneficial changes in an individual’s mood, behaviour, and overall functioning [49,50,51]. 

Hypnosis was found to be more effective than other psychological interventions [52] and, through a systematic review and meta-analysis of randomised controlled trials on the effect of hypnosis on anxiety, statistically significant results were found for its efficacy. It is considered that there is a ‘tremendous volume’ of research proving that hypnosis is very effective in the treatment of, and reduction in, anxiety [53]. 

Several studies using a mixed-model design in which hypnosis was compared with no treatment, attention, or standard care controls in reducing anxiety symptoms were identified. There is evidence from trials conducted around CBT and hypnotherapy that yielded a mean weighted effect size of 0.99 (*p* ≤ 001), demonstrating the average participant treated with hypnosis improved more than about 84% of control participants, and hypnosis was more effective in reducing anxiety when combined with other psychological interventions than when used as a stand-alone treatment [54].

A further study involved participants experiencing anxiety who were provided with a treatment choice labelled either “relaxation” or “hypnosis.” The actual treatment offerings were in fact identical for both groups. Participants who requested and received “hypnosis” showed greater improvement than the clients who received “relaxation” [49].

If psychological services such as hypnotherapy are helpful in reducing anxiety in clinical settings, the question was raised as to why OHS is not including this as a service.

### 3.2. Quantitative Survey

To determine OHS perceptions, a 46-item self-administered anonymous on-line survey was developed using QALTRICs. The questionnaire was developed using a participatory approach with a field of experts and representatives of the AIHS. The survey questions were co-designed to permit an evaluation of AIHS members’ perception regarding: the impact of employee anxiety on workplace performance; awareness of, and perceptions of hypnotherapy as a clinical therapeutic intervention; and receptivity towards hypnotherapy as an OHS service solution for employee anxiety. Questions on basic demographics, professional role and experience, and workforce size, were included, as were measures to gain a baseline understanding around beliefs about hypnotherapy and the perceived value of hypnotherapy. Questions were derived from the McConkey’s inventory [55,56].

An email invitation to participate in the survey was distributed by AIHS to 500 randomly selected members, and participants consented to participation on opening the survey.

The survey was completed by 191 AIHS members, giving a response rate of 38.2% (Table 1). A total of 136 participants completed all questions (81.7%) and data for this group were used for the analysis. The majority of respondents were male (56%), aged 50 years or older (60%), qualified OHS providers (79%), worked in organisations with 100 or more employees (74%), and had worked in their organisation for less than five years (55%).

The survey data were analysed using descriptive statistics using SPSS software (IBM SPSS and AMOS 28). Chi-square analysis was used to identify associations between parameters with an alpha set at 0.05.

#### OHS Perceptions Regarding Hypnotherapy for Anxiety Management

One hundred percent of participants agreed that employee anxiety impacts productivity in the workplace. A share of 65% indicated that employee psychological well-being and safety was a priority for their organisation (Figure 2). Approximately half of the respondents believed that hypnotherapy was a useful modality to manage psychological well-being and had a place in modern medicine; however, more than 40% were uncertain. 

With regard to hypnotherapy being a workplace OHS offering, only 20% believed that it had a place, and more than 60% were uncertain. 

Fewer than 7% of participants felt there was a clinical role for hypnotherapy, and this increased to 17.6% when considering its role in the workplace.

More younger participants (30–29 years) and older participants (60+ years) tended to have more favourable views towards a role for hypnotherapy in the workplace (*p* = 0.013), and there was a non-significant trend for males to have more favourable views compared to females (*p* = 0.069) (Table 2). Participants who agreed that there was a place for hypnotherapy in modern medicine were significantly more likely to agree that there was a place for hypnotherapy in workplace OHS.

There were no associations between the AIHS participant profile parameters and the response to the question of perceived barriers to the uptake of hypnotherapy of workplace OHS. Although not statistically significant, a greater proportion of participants from large organisations agreed that there were perceived barriers than those from smaller organisations. There was also a non-significant trend for a greater proportion of participants who felt that perceptions of hypnotherapy have been impacted by previous stereotypes or myths to agree that perceived barriers exist to the uptake of hypnotherapy in workplace. (Table 2).

### 3.3. Barriers to Adoption of Hypnotherapy

Nearly 85% of participants indicated that they perceived there would be barriers to the adoption of hypnotherapy for use by OHS services in their workplace (Figure 3). 

The main barriers given in the participants’ supplementary response included a lack of evidence and understanding of its benefits, credibility, potential cost, and perceptions. 

A share of 64% of participants agreed that perceptions of hypnotherapy have been impacted by previous stereotypes or myths.

## 4. Qualitative Focus Groups

AIHS members were invited to participate in a follow-up focus group session to develop a deeper understanding regarding the perceptions of and barriers to hypnotherapy. This discussion allowed for further information to be gathered and to determine if there was triangulation of the survey findings with the focus group information. The focus group facilitator encouraged open discussion in response to questions regarding beliefs about anxiety impacting individuals’ work productivity; hypnotherapy as a therapeutic option for anxiety; and perceived barriers to adoption of hypnotherapy by OHS workplace services. 

The format of the focus group used standardised, open-ended questions, with the same open-ended questions presented to all interviewees, and respondents were free to choose how to answer the question; they did not select “yes” or “no” or provide a numeric rating. 

### 4.1. Triangulation of the Data

Themes emerging from the focus group concurred with the quantitative data from the survey. These included barriers to implementation of hypnotherapy such as uncertainty around effectiveness (25% of participants), myths (21%), and, therefore, credibility (19%). 

Recognised themes emerging from the converged date were

Theme 1—Lack of Acceptance.

This refers to the feeling that participants had about how anxiety is either supported in the workplace or sabotaged. Comments shown below supported this theme: 


*“Senior staff are less likely to recognise a day needed for mental well-being over a day off for personal care. It is like they don’t understand the relationship my anxiety has with my output”.*



*“I think a lot of people just consider anxiety to be on the same level as stress. Stress is more temporary, and I would like others to know more about anxiety and accepted its place in our life”.*



*“COVID-19 really changed how I viewed anxiety. I guess I always thought of it as something weak people had or something that people needed to just get over. My workplace really hasn’t embraced how it impacted our daily lives and how to alleviate some of the issues”.*


Theme 2—Disclosure.

There is no obligation for workers to disclose information about their well-being. The context and perspective of how disclosing their health would impact their work can be impacted by a number of considerations, including their security around work performance, attitudes, and discrimination. Employees may have past experience of being discriminated against and this can impact how they communicate their well-being. 


*“I wouldn’t dare tell the people around me at work anything about my personal life. You don’t know who will share what story with who and what will end up happening”.*


Because of the pervasiveness of stigma, the prospect of disclosing anxiety concerns to an employer can be frightening, leading to many employees choosing to avoid disclosure altogether.

Disclosure has implications for both the employer and employee and relates to the culture of the workplace.

Theme 3—Unknowns.

Hypnotherapy as a treatment for anxiety was not a forthright discussion point for the participants. Hypnotherapy research outside of clinical studies often relies on anecdotal or aged information, and a relative lack of theory represents an impediment to the continued development of the scientific understanding of hypnosis, including its integration into broader models of human cognition [57,58]. 

### 4.2. Other Considerations

There are other perceived obstacles surrounding hypnotherapy in the workplace. These may include various misconceptions and fears surrounding hypnotherapy, stemming from the historical background and misuse of hypnosis by those commonly referred to as “showmen and charlatans” [59]. It may be necessary to highlight credentials, qualifications, and experience of the therapists involved to help alleviate this obstacle. 


*“Is a hypnotherapist truly qualified? I am not sure how they are certified, and I really don’t know much about how they would actually provide a solution for staff with anxiety”.*


An additional challenge may be that hypnotherapy research continues to be labelled “unscientific” in the broader research community [60,61]. However, there is some indication that the misconceptions about hypnosis may be softening due to general research progress.


*“Perhaps if someone had provided me with an understanding of hypnosis and how it all worked, I might have had included it in my toolbag”.*


Illuminating the field of hypnotherapy for organisational leaders and OHS providers who are seeking the best mental health support for employees under their duty of care may help improve both mental health and productivity.

The lack of reliable information regarding the discipline or field of hypnotherapy to be able to make informed workplace OHS decisions was a recurring theme affecting the receptivity towards hypnotherapy as an efficacious intervention in the workplace. 

### 4.3. Opportunities

During the focus group discussions, there was interest in and requests for workshops and provision of contemporary good quality evidence to address gaps in understanding regarding the potential importance and benefits of hypnotherapy for management of anxiety. There was a common view that the field and practice was not well communicated, and therefore not well understood, and, due to previous misinformation or stereotyping, hard to present as a credible offering within workplace discussions.

It was expressed that there was not always a clear understanding of the terminology around hypnosis and hypnotherapy, and confusion around the therapy, and further clarity around the differences or similarities would help address this issue.

To address misinformation and gaps in information, increasing the number of empirical studies should help mediate some flawed perceptions and help to overcome barriers.

As a conclusion, there is a theory that can be proposed that the low level of awareness or receptivity within OHS is related to access to information, misinformation, or perceptions that are incorrect based on historical stereotypes or myths surrounding the practice of hypnotherapy as assessed from the study.

### 4.4. Limitations

There is a risk of bias in non-experimental research that employs collected data. Due to the problem of bias, the results may be different from the truth or true values. All observational studies have attached biases, and bias is a limitation with regard to research. The current study may be subject to selection bias as it involved the selection of participants for a treatment program, which may result in distorted outcomes. Selection bias is found frequently in program interventions where participants are self-selected into hypnotherapy programs [62]; therefore, the results are likely to be distorted, leading to generalised estimates since the outcomes do not guarantee the real effect of interventions. For instance, in this study, the effect of hypnotherapy on anxiety in the organisations [30] may apply more to those employees who are more in need of the assistance. 

With regard to voluntary sampling, the participants often have a strong interest in the main topic of the survey. In this case, subjects who participate in the self-administered questionnaire may not be as likely to be emotionally unstable as are those in the clinical populations, precisely because the latter are experiencing some sort of difficulty. This may result in those who are less anxious choosing to participate at a higher rate than those who may essentially need treatment more.

## 5. Discussion

The results presented here are only an overview of the main results collected within a larger representative survey of those involved in OHS delivery within workplaces. Further analysis is underway, including factor and multivariate analyses, to better profile the opinions and perceptions to help target education and resources.

Because empirical studies are limited in terms of quality and volume [63] in the workplace setting, and hypnotherapy research outside of clinical studies often relies on anecdotal information, there is a relative lack of theory. This represents an impediment to the continued development of the scientific understanding of hypnosis [14,64]. 

Adding to, and improving on, empirical studies will be critical if the field is to be developed for further use outside clinical application [39]. 

The barriers to acceptance in hypnotherapy that were discovered through this study may be mitigated by addressing the misconceptions around the history of the practice, misinformation, or the lack of information regarding its efficacy, and by undertaking high quality workplace research [37,59].

Illuminating the field of hypnotherapy for organisational leaders and OHS providers who are seeking the best mental health support for employees under their duty of care is a strong motivation for pursuing this research, as is the lack of OHS empirical and management studies in the area [24,27,45].

There is academic, industry, and social implications and significance for this research, as it can inform research institutions and OHS providers who are assessing workplace anxiety treatments; individual employees impacted by anxiety; and these employees’ families, their colleagues, and the workplace communities they belong to [13,65].

The potential for organisations to address a growing issue around the issue of anxiety management by adopting practices that clinicians have trialled may be a relatively easy and inexpensive way to provide a wide range of beneficial “side effects” for employees, such as an increased sense of well-being, and very few negative side effects [16,29,31,32,34].

With growing indications that hypnotherapy is a successful method of dealing with anxiety, there should be increased interest in conducting more research and developing helpful interventions for OHS practitioners in this field.

## 6. Conclusions

Although hypnotherapy is increasingly practiced by conventional medical practitioners and is becoming a popular and widely known complementary medical therapy, within the cohort of OHS professionals surveyed there remains apprehension, scepticism, and suspicion around the subject. 

The data from this mixed methods study confirms that there is a high degree of uncertainty regarding whether the use of hypnotherapy outside clinical settings and in workplaces would be supported. 

The barriers to receptivity appear to be due to a lack of information and knowledge, and misconceptions about the practice. These are sometimes affected by the media, which may treat the practice as a field of entertainment or mystique. 

These conclusions can help inform hypnotherapy governing bodies and clinicians, with the outcome of improving the communication and presentation of evidence from the field to a wider audience.

## Figures and Tables

**Figure 1 ijerph-19-07735-f001:**
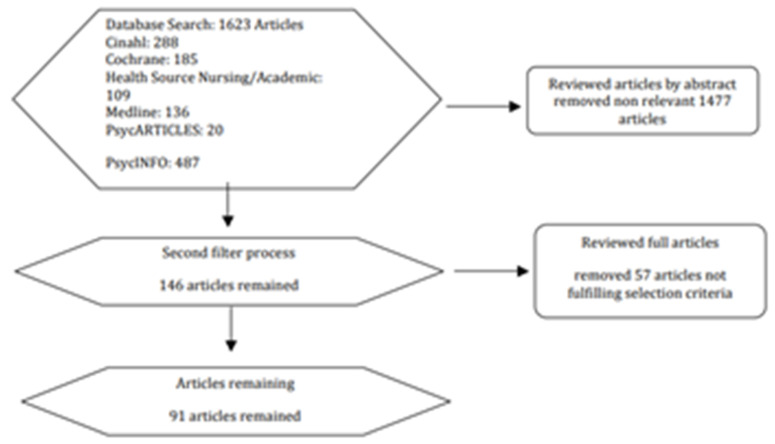
Literature review filtering process.

**Figure 2 ijerph-19-07735-f002:**
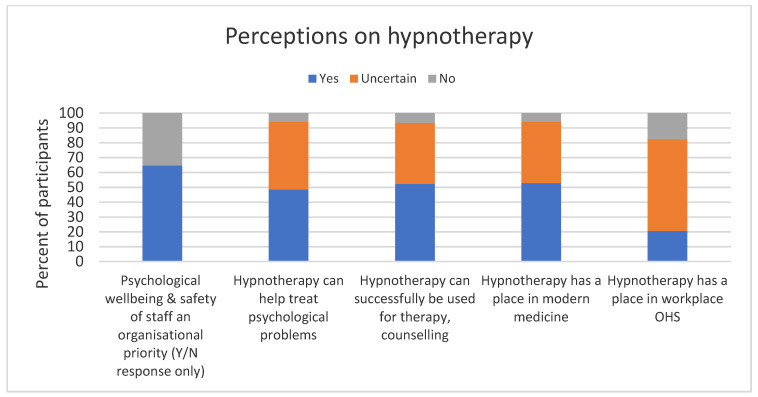
Perceptions regarding hypnotherapy.

**Figure 3 ijerph-19-07735-f003:**
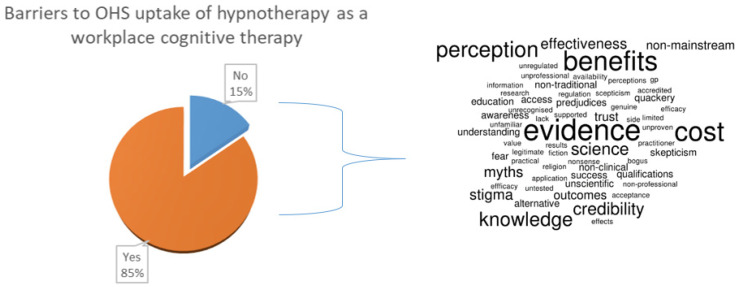
Perceived barriers towards hypnotherapy as an OHS offering.

**Table 1 ijerph-19-07735-t001:** AIHS participant profile for the survey and focus group.

	Survey *n* (%)	Focus Group *n*
N	136	12
Female: male	60:76 (44.1:55.9)	11:1
Age group (years)		
- 18–29	2 (1.5)	1
- 30–39	23 (16.9)	6
- 40–49	29 (21.3)	4
- 50–59	52 (38.2)	1
- 60+	30 (22.1)	0
Role in organisation		
- Executive, administrator, or senior manager	17 (12.5)	1
- Line manager with OHS responsibility	11 (8.1)	5
- Qualified OHS provider	108 (79.4)	6
Number of employees		
- Less than 10	18 (13.2)	3
- 10–99	17 (12.5)	8
- Over 100	101 (74.3)	1
Time in organisation (years)		
- Less than 5	75 (55.2)	1
- 5–10	26 (19.1)	8
- 10+	35 (25.7)	3

**Table 2 ijerph-19-07735-t002:** Associations between AIHS participant profile and perceptions and barriers to OHS hypnotherapy.

	Place in the Workplace OHS*p* Value	Barriers to OHS Uptake*p* Value
Sex (F:M)	0.069	0.200
Age group	0.013	0.605
Role in organisation	0.301	0.327
Number of employees	0.142	0.066
Time in organisation	0.551	0.135
Place in modern medicine	0.001	0.469
Myths	0.321	0.069
Place in workplace OHS		0.207
Barriers to uptake	0.207

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
