# Peer review of "Occupational Health and Safety Receptivity towards Clinical Innovations That Can Benefit Workplace Mental Health Programs: Anxiety and Hypnotherapy Trends"

_ijerph, 2022, doi:10.3390/ijerph19137735_

Round 1

Reviewer 1 Report

This manuscript evaluates the role of Hypnotherapy as a helpful cognitive therapy used in clinical practice to reduce day to day anxiety especially managing workplace stressors. Author has given a good understanding about anxiety and how it affects the performance of an individual at workplace. It also reflects the role of occupation health and safety (OHS ) in managing the response of an individual to day to day stressors using hypnosis and hypnotherapy.

This manuscript covers different aspects of hypnotherapy in managing day to day stress and anxiety. However, in order to recommend this for editorial decision author needs to clarify following comments:

1) What measures can be taken to remove misinformation regarding hypnotherapy and how it can be made more readily available to individuals who are willing to take it.

2) How to make Hypnotherapy more effective to help individuals deal with day to day stressors especially workspace stressors.

Author Response

Dear reviewer, thank you for your generous and valuable input and comments. They are greatly appreciated.

in reply to your suggestions of including

1.measures to mitigate against misinformation and also,

2. ways in which to help translate hypnotherapy into work place settings.

Good insight ! thank you. I will work on that for the resubmission. I agree it will strengthen the paper.

Again, my appreciation for you time and review.

Reviewer 2 Report

The article is very interesting and well thought out, but a number of improvements are suggested.

The summary focuses mainly on the introduction and the sections on objectives, methodology, results and conclusion are not clear. The methodology section is not very clearly laid out and organised, it is recommended that it be reorganised. The quality of the figures is rather low. The discussion is too brief and there are few quotations from recent years. A restructuring is recommended, following the guidelines of the review process. It is also recommended that the title of the article be reconsidered to indicate more clearly the objective of the article.

Author Response

Dear reviewer, thank you for your time and review of the paper. It is greatly appreciated.

In light of your suggestions I will certainly look at editing the section on methodology, results and conclusion to make them more clear and laid out in a different fashion. I will also reconsider the title as you suggest.

I know that your time is precious so I appreciate your review and the input will be helpful to land on a finished paper for resubmission.

Reviewer 3 Report

In this manuscript authors presentes a study conducted in Australia to evaluate if hypnotherapy is considered a helpful cognitive therapy used in clinical practice, how do OHS professionals view the use of hypnotherapy as an alternative service to deal with anxiety for employees, as well as possible barriers that may need to be overcome if it was to be considered a workplace offering.

The argument of the manuscript it is original, but the text presents serious shortcomings: a specific paragraph on the purpose of the study is missing, the materials and methods section and the detailed explanation of the literature review strategy adopted are missing, a separate section relating to the results that emerged is missing, there is a partial description of the literature studies detected, discussion section lacks contents on the implications this study could have on national policies and what the paper adds to scientific literature. It is necessary that the authors investigate these aspects in depth, highlighting the areas in which these results could have healthcare implications. 

In general, the English in the paper can be understood.

The Major Essential Revisions include: 

-       Authors should include a specific paragraph related to the purpose of the study

-       Lines 36-38: This concept should be expanded further

-       Authors should add a paragraph relating to materials and methods specifying the search strategy adopted (e.g. PICO strategy)

-       Lines 178-185: this part of the text should be moved to the introductory part paying attention to possible repetitions with previously expressed concepts

-       From line 195: Authors should reserve a specific section for manuscript results

-       Line 199: the concept "tremendous value" assumes that several scientific papers have been found that refer to hypnosis as an effective method in the treatment and reduction of anxiety, but only one bibliographic reference has been reported. Authors should expand on this part.

-       It would be useful to create a table indicating the studies analyzed and the main results

-       The studies described in the text are far fewer than the 91 studies analyzed. Summarize and categorize the main results that emerged from the 91 studies

-       Lines 229-230: It is not clear whether the figure refers to a study already carried out or which will be carried out in the future. Tense is future

-       The figure presented is not easy to understand. Authors should describe in the text the variables considered and the main results that emerged

-       A section dedicated to discussion is missing. Authors should add it

-       A section dedicated to the contributions of the authors is missing. Authors should add it

-       A section relating to funding and conflict of interest is missing. Authors should add it

-       Discuss potential limitations of the study within a specific paragraph, taking into account critical points, potential bias or imprecision. 

Author Response

Dear reviewer, thank you for the time and patience to review the submitted article. Your input and suggestions are invaluable and will improve the impact of the research. I will indeed look at improving the sections that can shed more ligh on the implications this study could have on national policies and how the paper will add to scientific literature.

I know your time is precious and I really appreciate the great guidance you have given.

Many thanks.

Reviewer 4 Report

 This study was conducted to identify barriers and evaluate potential acceptance in the workplace for the use of hypnotherapy for reducing anxiety.  In Australia, occupational health and safety (OHS) practitioners are among those responsible for implementing and managing workplace programs for addressing worker anxiety.  According to the literature review, anxiety can have a variety of effects in the workplace, including reducing productivity and work quality, disrupting relationships with co-workers and superiors.  The lack of research on precursors to anxiety, outcomes of anxiety on work performance, and the ability of OHS practitioners and systems to address anxiety and other mental health conditions are also discussed.  In addition, several studies demonstrating the effectiveness of hypnosis for reducing anxiety were cited. 

The investigator then discussed the study that was the focus of this paper, which included a survey and focus groups and/or interviews held to develop ideas for recommendations for addressing anxiety in the workplace.  As a result of the survey and interviews, barriers to acceptance of hypnosis as a treatment for anxiety were identified, and suggestions for overcoming those barriers and ideas for future research were suggested.

Several questions and information gaps were identified and are listed below.  It would be helpful if these could be addressed in the paper. 

1.      As background information, it would be useful to provide the requirements in Australia for anxiety to be accepted as a mental health work claim.

2.      How common is it for occupational safety and health practitioners to address anxiety through programs or other means in the workplace?  Is this currently being done in Australia?  Have workplace programs (similar to workplace safety programs) for preventing and treating anxiety been developed and are they available for companies and OHS practitioners to use?

3.      It would be helpful to include information on the scope of the problem of anxiety in the workplace, such as number of people affected annually, estimated reduction in productivity in workplaces as a result of anxiety, etc.  Also, can the author clarify what the percentages in lines 73 – 75 refer to?

4.      A definition, description, or examples of hypnotherapy and how it is related to hypnosis would be helpful.  While some information is provided in lines 190 – 193, a description of the steps that are taken in hypnotizing a worker or how the OHS practitioner and worker interact during a session would be informative. 

5.      In lines 195 – 200 and 201 - 208, it was noted that hypnotherapy is effective for reducing anxiety.  How was anxiety measured? What are the treatments against which hypnotherapy was compared?  What were the workplace outcomes associated with anxiety reduction?  Increased productivity, etc.?

6.      What was the point of including the study by Frankel and Zamansky (1978) in lines 214 – 216.  Study findings suggest the need for blinding subjects to their intervention/control group and the power of the placebo effect, but beyond that, the reason it was included in the literature review is not clear.

7.      The correlations and differences that were tested in the statistical analyses described on page 6 are unclear.  Can additional information be provided?

8.      What sectors and company sizes were participants from who completed the survey in the current study?  Was the Qualtrics survey developed specifically for this study?  Were the reliability and validity of the survey evaluated prior to use in the current study?

9.      How were focus group members and interview participants in the current study selected?  How many participants were there?   What sectors and company sizes did they represent?

10.  In line 318, self-hypnosis was suggested as a treatment for many health conditions including anxiety.  This was the first mention of self-hypnosis.  Does the information discussed in the paper on hypnotherapy apply to self-hypnosis also?  It would be useful to introduce the concept of self-hypnosis much earlier in the paper and provide a description of it and its use.  Introducing it at the end of the paper as a recommendation is confusing since it was not discussed earlier.

Author Response

Dear reviewer, thank you so much for your time and patience in reviewing the paper. You have raised 10 really helpful points which will be used to edit the paper ready for resubmission. The edits will include

Background information around the requirements in Australia for anxiety to be accepted as a mental health work claim

How common it is for occupational safety and health practitioners to address anxiety through programs and the types of workplace programs they use.

More information on the scope of the problem of anxiety in the workplace with statistics and clarification of the percentages in lines 73 – 75.

A clear definition of hypnotherapy, its parellel and differences to hypnosis as well as information is provided in lines 190 – 193, a description of how the OHS practitioner and worker interact to allow for this practice to occur. 

An explanation of the difference between diagnosed anxiety and feelings of anxiousness will be provided to ensure there is no confusion about the measurement of anxiety being an issue in this paper.

Provision of further information regarding the correlations and differences tested in the statistical analyses described on page 6.

Fuller descriptions on the sectors that participants were drawn from.

The Qualtrics survey was developed specifically for this study and evaluated by the HREC protocols in Australia as well as several Australian university research panels.

Focus group members and interview participants were invited through their organisations and able to self select into the research. I will detail that further in the paper.

The re edited paper will provide more detail on participant numbers

To remove confusion I will remove the In line 318 the concept of self-hypnosis.

Again I really appreciate your great insights and valuable input, it will be put to good use and that is the best thank you I can hope to give. However, a BIG thank you for your time and assistance. It is appreciated.

Round 2

Reviewer 1 Report

Author has made all the suggested changes in the manuscript. I highly recommend this manuscript for publishing.

Reviewer 2 Report

Congratulations to the authors for their work in improving the article. The work presented may be of interest to researchers in this scientific field.

Reviewer 3 Report

The authors answered comprehensively to requests for review